# MLLM-Pruner: Efficient Activation-aware Pruning for Multimodal LLMs

## Abstract

Multimodal large language models (MLLMs) have demonstrated impressive performance across a wide range of vision-language tasks. However, the increasing scale of these models leads to significant challenges in deployment costs. Post-training pruning emerges as an effective compression technique to address these challenges. Recent pruning studies on large language models (LLMs) has shown that activation-aware pruning strategies that combine weight magnitude with the $\ell_2$-norm of input activations can achieve superior performance. Nevertheless, directly applying these approaches to MLLMs often leads to substantial performance degradation. This is because the $\ell_2$-norm assumes all activations contribute equally, while in MLLMs, visual and textual tokens exhibit divergent activation patterns. Moreover, textual-only calibration datasets used in LLM pruning are inadequate for capturing modality-specific dependencies, which further limits their ability to evaluate the importance of weight. In this paper, we propose MLLM-Pruner, a novel activation-aware pruning framework specifically tailored for MLLMs. To address these issues, MLLM-Pruner introduces two key innovations: (1) we construct a representative multimodal calibration dataset comprising general-domain text, instruction tuning, and visual instruction tuning data to comprehensively preserve language generation, instruction-following, and visual reasoning abilities for MLLMs. (2) we design a modality-sensitive importance estimation metric that leverages the Singular Value Decomposition (SVD) of attention distributions to reweight the input activations, effectively captures the activation contribution across modalities and reduces the pruning error. Our MLLM-Pruner does not rely on expensive iterative reconstruction and re-training process. Extensive experiments on LLaVA-based MLLMs across various benchmarks demonstrate that MLLM-Pruner consistently outperforms state-of-the-art pruning methods while maintaining efficient compression. Our code, model weights, and multimodal calibration dataset will be made publicly available upon publication.

## 1 Introduction

Large language models (LLMs) (OpenAI, 2023; Touvron et al., 2023; Xue et al., 2020) have demonstrated impressive zero-shot abilities across different open-ended tasks, and Multimodal large language models (MLLMs) (Liu et al., 2023b; 2024a; Lin et al., 2023; Bai et al., 2023; Wang et al., 2024) extend LLMs with visual understanding capabilities. However, both LLMs and MLLMs typically contain billions of parameters, making practical deployment challenging due to their size and computational demands. To address this challenge, various model compression techniques have been proposed to reduce model size while preserving capability, including model quantization (Dettmers et al., 2022; Lin et al., 2024; Frantar et al., 2022), knowledge distillation (Hinton et al., 2015; Gu et al., 2023), and pruning (Han et al., 2015; He et al., 2017; Wang et al., 2019b), etc. Among them, pruning has emerged as an effective solution, as it removes redundant parameters, induces sparsity for computational acceleration, and requires no extensive retraining process. In this paper, we focus on post-training unstructured pruning for MLLMs, a training-free method that reduces model size without sacrificing its strong visual understanding potential.

Recently, a variety of conventional pruning methods (Mallya & Lazebnik, 2018; Molchanov et al., 2019; Frantar & Alistarh, 2022) have applied for LLMs. For example, SparseGPT (Frantar & Alistarh, 2023) achieves impressive results on LLMs through iterative optimization. However,

the iterative process requires computing the inverse of the second-order Hessian matrix for the entire model weights, which incurs prohibitive computational and memory overhead. In contrast, magnitude-based methods (Han et al., 2015; Chen & Zhao, 2018; Sun et al., 2023) are computationally efficient, which assume that weights with large magnitudes are informative, and prune the uninformative weights in a single iteration without expensive update problems. Wanda (Sun et al., 2023) introduces an activation-aware pruning metric, which defines the importance of each weight as the product of its magnitude and the $\ell_2$-norm of the corresponding input activation. This simple metric has been shown to preserve the LLM performance even under high compression ratios.

Despite these advantages, the potential of activation-aware pruning for MLLM compression has not been fully explored. Unlike LLMs, MLLMs exhibit architectural complexity and involve non-uniform activation behaviors across modalities, where input sequences typically consist of image embeddings, textual instructions, and answers, each playing distinct roles in activation behavior and information flow. However, the conventional $\ell_2$-norm of the input activations treats all tokens as equally important, which limits its effectiveness for MLLMs and often leads to performance degradation. Furthermore, existing pruning approaches typically estimate weight importance using a small calibration dataset such as C4 (Raffel et al., 2020), a general-domain corpus typically used for the LLM pre-training process. However, MLLMs exhibit significant activation differences between visual and textual modalities; moreover, prior studies (Chen et al., 2024a) have also observed that early layers in MLLMs tend to assign more attention to the visual tokens, whereas the C4 dataset lacks visual content and the corresponding attention patterns. These cross-modal disparities pose significant challenges for designing a unified calibration strategy suitable for MLLM pruning.

In this paper, we propose MLLM-Pruner, an activation-aware post-training pruning framework tailored for multimodal large language models (MLLMs). Our approach addresses the key challenges of MLLM pruning from two perspectives: (1) Multimodal Calibration Dataset. We design a calibration strategy specifically for MLLMs, ensuring that the calibration data are both general and representative, which enables more effective weight-importance evaluation. (2) A Modality-sensitive Importance Estimation Metric: a novel weight importance estimation method, which not only considers the magnitude of the weights and the input activations, but also introduces modality-sensitive contribution scores for activation reweighting. This design explicitly captures cross-modal activation divergence, enabling MLLMs to more precisely identify informative weights across different modalities. We compare our MLLM-Pruner with both the state-of-the-art activation-aware pruning method and the iterative optimization-based method. We conduct extensive experiments towards the LLaVA-NeXT (Liu et al., 2024a) 7B and 13B across a variety of MLLM evaluation benchmarks. Our results demonstrate that MLLM-Pruner consistently outperforms the state-of-the-art pruning methods, achieving superior trade-offs between pruning efficiency and model performance. We summarize our contributions as follows:

- We propose MLLM-Pruner, a novel activation-aware post-training pruning framework for compressing MLLMs. MLLM-Pruner introduces a modality-sensitive importance estimation metric that explicitly accounts for multimodal activation divergence, enabling more accurate weight importance estimation for MLLM pruning.

- We construct a multimodal calibration dataset that provides informative and representative statistics for pruning, effectively preserving the continuation, generation, and visual understanding abilities of MLLMs through our hybrid data calibration strategy.

- We validate our MLLM-Pruner on LLaVA-NeXT 7B and 13B across MLLM evaluation benchmarks, achieving 3.3% and 1.4% relative improvements on average performance compared with the baseline. We provide extensive analyses and ablation studies that help to understand the challenges of MLLM pruning and the strengths of our proposed methods.

## 2 RELATED WORK

**Post-Training Pruning** compresses a well-optimized model by retaining only critical parameters while maintaining the performance (He & Xiao, 2023; Cheng et al., 2024), which is commonly divided into structured and unstructured forms depending on the pruning granularity. The structured method (McCarley et al., 2019; Kwon et al., 2022; Ma et al., 2023) is hardware-friendly and enables efficient inference acceleration, whereas unstructured pruning (Dong et al., 2017; Lee et al., 2019;

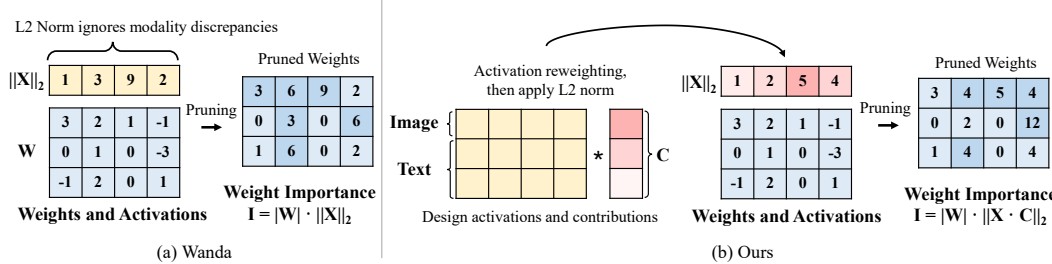

Figure 1: The illustration of our proposed MLLM-Pruner. Compared with Wanda (Sun et al., 2023), which prunes LLMs by weights and activations, we introduce a novel weight importance estimation metric for MLLMs. By designing representative activations and their contributions, we reweight the input activations to capture the modality discrepancies, enabling a more reliable informative estimation of the weights in MLLMs.

Park et al., 2020) based on sparse matrix computation schemes can better preserve the model performance. Existing approaches often formulate it as a layer-wise optimization problem, with minimizing compression loss as the objective, and several criteria have been introduced to remove the structure or parameters, such as magnitude-based methods (Han et al., 2015) and gradient-based (Hou et al., 2020; Kurtic et al., 2022; Wang et al., 2019a) estimation.

**MLLM Pruning.** Multimodal large language models (MLLMs) (Li et al., 2021; Liu et al., 2023b; 2024a; Bai et al., 2023; Wang et al., 2024) have gained much attention due to their strong capabilities across different vision-language tasks, which highlights the need to reduce parameter sizes for deployment across diverse scenarios. Recently, there has been an increasing focus on pruning the large language models (LLMs) themselves. SparseGPT (Frantar & Alistarh, 2023) employs a sparse regression solver through Hessian reconstruction based on classic Optimal Brain Surgeon update (Hassibi et al., 1993; Frantar & Alistarh, 2022), where the reverse of the Hessian reconstruction process is computation cost when minimizing compression loss. Wanda (Sun et al., 2023) proposes a computation-friendly magnitude-based approach, which evaluates the importance of weights by taking the input activations into consideration. However, the aforementioned approach faces challenges when applied to MLLMs due to the big modality discrepancies. In addition, post-training pruning is usually conducted with a limited calibration dataset (e.g., C4 (Raffel et al., 2020), WikiText (Merity et al., 2016)), whereas the feasibility of applying the same calibration strategy to MLLMs has not been sufficiently explored. In this paper, we focus on magnitude-based unstructured pruning, and rethinking the post-training pruning paradigm for MLLMs.

## 3 METHODS

In this section, we present MLLM-Pruner, a novel activation-aware pruning framework for multimodal large language models (MLLMs). An overview of the framework is shown in Fig. 1. Section 3.1 reviews the foundations of magnitude-based pruning methods and the attention mechanisms in MLLMs, while Section 3.2 describes the details of our proposed approach. which introduces a new multimodal pruning metric that explicitly accounts for the cross-modal activation divergence and their contributions, enabling more accurate re-evaluation of weight importance in MLLMs.

### 3.1 PRELIMINARIES.

#### 3.1.1 MAGNITUDE-BASED MODEL PRUNING.

We start by introducing the canonical formulation of layer-wise pruning. Existing methods formulate pruning as an optimization problem (Hubara et al., 2021; Frantar & Alistarh, 2023) by selecting a sparsity mask $\mathbf{M}$ for weight matrix $\mathbf{W}$, formulated as:

$$\min_{\mathbf{M}} \ \|\mathbf{W}\mathbf{X} - (\mathbf{M} \odot \mathbf{W})\mathbf{X}\|_2^2 \tag{1}$$

Magnitude-based pruning (Han et al., 2015) constructs the sparsity mask $\mathbf{M}$ by ranking weight elements according to their absolute values, where the importance score $I_{ij} = |W_{ij}|$. The sparsity

mask $\mathbf{M}$ is then constructed by setting $\mathbf{M}_{ij} = 0$ if $I_{ij} \leq \tau$, where $\tau$ is the threshold determined by the target sparsity ratio. Wanda (Sun et al., 2023) improves this by introducing an activation-aware pruning metric, which redefines the importance score as $I_{ij} = |W_{ij}| \cdot \|\mathbf{X}_j\|_2$, where $\|\mathbf{X}_j\|_2$ is the $\ell_2$-norm of the $j$th feature aggregated across N tokens. However, this criterion implicitly treats all tokens equally, which is suboptimal for MLLMs with high variance of input activations.

### 3.1.2 ATTENTION IN MLLMs

Due to the modality heterogeneity in MLLMs, our target is to reweight the activation for magnitude-based pruning. One key insight is that the attention matrix naturally reflects modality discrepancies of the input activations. Given an input activation sequence $\mathbf{X}$, the multi-head attention matrix of every layer is defined as:

$$\mathbf{A} = \text{softmax}\left(\frac{\mathbf{Q}\mathbf{K}^\top}{\sqrt{d_K}}\right) \tag{2}$$

where $\mathbf{A} \in \mathbb{R}^{H \times (N+M) \times (N+M)}$, $N$ and $M$ are the numbers of visual and textual tokens, respectively, $H$ is the number of attention heads, and $d_K$ is the scaling factor.

However, the attention mechanism in MLLMs is unidirectional, enforcing the information flow from earlier tokens to subsequent ones. Consequently, the information of subsequent tokens remains "invisible" to preceding tokens, leading to biased and non-uniform importance estimation. To obtain a more reliable measure of activation importance, we design a complementary metric to capture the global information of the activations.

### 3.2 MLLM-PRUNER

We propose MLLM-Pruner to address the aforementioned challenges through three steps: (1) constructs a crafted multimodal calibration dataset for MLLMs. (2) introduce a novel modality-sensitive weight importance estimation metric, which reweights the activation according to the activation contribution to capture the data-level and modality-level variance. (3) aggregates the importance score across different types of calibration datasets for the final pruning.

### 3.2.1 MULTIMODAL CALIBRATION DATASET

To preserve the continuation, generation, and visual understanding abilities of MLLMs, we construct a multimodal calibration dataset comprising three complementary sources: the general-domain corpus C4, Instruction Tuning data (IT), and Visual Instruction Tuning data (VIT). Specifically, the input forms are $\mathcal{D}_{\text{C4}} = \{X_t\}$, $\mathcal{D}_{\text{IT}} = \{X_{\text{ins}}, X_{\text{ans}}\}$, and $\mathcal{D}_{\text{VIT}} = \{X_v, X_{\text{ins}}, X_{\text{ans}}\}$, where $X_t$, $X_{\text{ins}}$, $X_{\text{ans}}$, and $X_v$ represent the pure text sequences, instructions, answers, and visual tokens, respectively. Here, $\mathcal{D}_{\text{C4}}$ amis to preserve continuation and generation ability, $\mathcal{D}_{\text{IT}}$ enhances instruction following, and $\mathcal{D}_{\text{VIT}}$ strengthens multimodal alignment and visual reasoning. Together, these datasets form a balanced calibration set for effective MLLM pruning.

### 3.2.2 MODALITY-SENSITIVE WEIGHT IMPORTANCE ESTIMATION

For each calibration dataset $d \in \mathcal{D}$, where $\mathcal{D} = \{\mathcal{D}_{\text{C4}}, \mathcal{D}_{\text{IT}}, \mathcal{D}_{\text{VIT}}\}$, we deine their input activations as $\mathbf{X} \in \mathbb{R}^{(N+M) \times C_{\text{in}}}$, where $N$ and $M$ are the numbers of visual and textual tokens, respectively. For calibration dataset $\mathcal{D}_{\text{C4}}$ and $\mathcal{D}_{\text{IT}}$ that without image token insert, $N = 0$. Given a linear layer $\mathbf{W} \in \mathbb{R}^{C_{\text{out}} \times C_{\text{in}}}$ in MLLMs, we introduce two complementary importance estimates for the corresponding input activations to better evaluate weight significance: **Attention-based Contribution**, which measures the averaged attention distribution over all tokens and captures activation importance along the causal direction of information flow; and **SVD-based Contribution**, which quantifies token importance through singular value decomposition of the attention matrix, providing a uniform information estimate that mitigates the biases. We obtain the final activation contribution score by combining them for each dataset $d$, which guides the weight importance evaluation in the pruning process.

**Attention-based Contribution.** To measure the importance of all tokens, we firstobtain the attention scores of MLLMs. Let $\bar{\mathbf{A}}^l \in \mathbb{R}^{(N+M) \times (N+M)}$ denote the attention matrix averaged over

multi-head in the $l$-th layer, We define the averaged attention score of token $j$ as:

$$a_j^l = \frac{1}{N+M} \sum_{i=1}^{N+M} \bar{A}_{ij}^l, \quad j = 1, \ldots, N+M, \tag{3}$$

**SVD-based Contribution.** For averaed attention matrix $\bar{\mathbf{A}}^l \in \mathbb{R}^{(N+M)\times(N+M)}$ at layer $l$, we then apply Singular Value Decomposition (SVD) (Eckart & Young, 1936; Golub et al., 1987) for a low-rank decomposition:

$$\bar{\mathbf{A}}^l = \mathbf{U}^l \mathbf{\Sigma}^l (\mathbf{V}^l)^\top, \tag{4}$$

where $\mathbf{U}^l, \mathbf{V}^l \in \mathbb{R}^{(N+M)\times(N+M)}$ are the left and right singular vector matrices, and $\mathbf{\Sigma}^l = \mathrm{diag}(\sigma_1^l, \ldots, \sigma_{N+M}^l)$ is the singular values of attention matrix $\bar{\mathbf{A}}^l$. The SVD-based contribution of token $j$ at layer $l$ is obtained by summing its loadings $U_{ji}^l$ across all singular directions, each weighted by the corresponding singular value $\sigma_i^l$, denoted as:

$$s_j^l = \sum_{i=1}^{N+M} \left| U_{ji}^l \, \sigma_i^l \right|, \quad j = 1, \ldots, N+M, \tag{5}$$

**Activation Reweighting.** For all tokens, we obtain their attention-based and SVD-based contribution in the $l$-th layer, respectively, denoted as:

$$\mathbf{a}^l = (a_1^l, a_2^l, \ldots, a_{N+M}^l) \in \mathbb{R}^{N+M}, \qquad \mathbf{s}^l = (s_1^l, s_2^l, \ldots, s_{N+M}^l) \in \mathbb{R}^{N+M} \tag{6}$$

We apply min-max normalization to place the two contributions on a comparable scale:

$$\hat{\mathbf{a}}^l = \mathrm{Norm}(\mathbf{a}^l), \qquad \hat{\mathbf{s}}^l = \mathrm{Norm}(\mathbf{s}^l), \tag{7}$$

The final input activation contribution is obtained by:

$$\mathbf{C}^l = \beta \, \hat{\mathbf{a}}^l + (1 - \beta) \, \hat{\mathbf{s}}^l, \qquad \beta \in [0, 1], \tag{8}$$

where $\beta$ controls the trade-off between the attention-based and SVD-based contributions. For each calibration dataset $d$, this final contribution score is then applied to reweight the input activations:

$$\tilde{\mathbf{X}}^{l,d} = \mathbf{X}^{l,d} \cdot \mathbf{C}^{l,d}, \tag{9}$$

where $\mathbf{X}^{l,d}$ is the original activation of the $l$-th layer for dataset $d$, $\mathbf{C}^{l,d} \in \mathbb{R}^{N+M}$ is the corresponding contribution score, and $\cdot$ denotes element-wise multiplication along the token dimension.

### 3.2.3 CROSS-DATASET AGGREGATION.

The reweighted activation $\tilde{\mathbf{X}}^{l,d}$ is computed separately for each calibration dataset $d \in \mathcal{D}$, where $\mathcal{D} = \{\mathcal{D}_{\mathrm{C4}}, \mathcal{D}_{\mathrm{IT}}, \mathcal{D}_{\mathrm{VIT}}\}$. Within each dataset $d$, we follow Wanda (Sun et al., 2023) and employ a sliding average to accumulate stable statistics of the reweighted activation:

$$\mathbf{S}_{(t)}^{l,d} = \frac{n}{n+m} \mathbf{S}_{(t-1)}^{l,d} + \frac{1}{n+m} \left\| \tilde{\mathbf{X}}_{(t)}^{l,d} \right\|_2^2, \tag{10}$$

where $n$ is the number of previously processed samples, $m$ is the number of current samples, and $\tilde{\mathbf{X}}_{(t)}^{l,d}$ is the current reweighted input activation of the $l$-th layer for calibration dataset $d$.

After obtaining the accumulate score $\mathbf{S}^{l,d} \in \mathbb{R}^{C_{\mathrm{in}}}$ of each calibration dataset $d$, we then aggregate across whole datasets $D$ given their sample-ratio $\alpha_d$, then the score for whole samples is:

$$\mathbf{S}^{l,D} = \sum_{d \in \mathcal{D}} \alpha_d \mathbf{S}^{l,d}, \qquad \sum_{d \in \mathcal{D}} \alpha_d = 1, \tag{11}$$

where $\alpha_d$ is the proportion of dataset $d$ relative to the total calibration sample size, and finally, the weight importance estimation metric of the whole calibration dataset $\mathcal{D}$ for pruning is defined as

$$I_{ij}^l = \left| W_{ij}^l \right| \cdot \sqrt{\mathbf{S}_j^{l,D}}. \tag{12}$$

This weight importance estimation metric serves as the final criterion to determine which weights are informative and need to be preserved or removed during pruning, ensuring that both modality-specific and cross-dataset information are properly considered.

# 4 EXPERIMENTS

## 4.1 EXPERIMENTAL SETTINGS

**Models and Evaluation.** We evaluate our method on the LLaVA architectures, which demonstrate superior performance among open-source Multimodal Large Language Models (MLLMs). Specifically, we focus on LLaVA-NeXT (Liu et al., 2024a) 7B and 13B. For evaluation, different from the large language models (LLMs) that use perplexity as an evaluation metric after pruning (Dettmers & Zettlemoyer, 2023), we measure the zero-shot performance of the pruned MLLMs on various vision-language benchmarks, including POPE (Li et al., 2023), ScienceQA (Lu et al., 2022), TextVQA (Singh et al., 2019), MME (inscluding Perceotion and Cognition) (Fu et al., 2023), GQA (Hudson & Manning, 2019), MMBench (Liu et al., 2024b), MMVet (Yu et al., 2023), VizWiz (Bigham et al., 2010), DocVQA Mathew et al. (2021), OCRBench Liu et al. (2024c), MM-Star (Chen et al., 2024b) .etc. We compare our pruning methods with three superior baselines: Magnitude-based pruning (Han et al., 2015), iterative optimization-based SparseGPT (Frantar & Alistarh, 2023), and activation-aware Wanda (Sun et al., 2023), under the same settings for a fair comparison. We report both the performance of pruned models and the full (dense) models. To quantify the overall effectiveness, we further compute the average relative performance, denoted as Avg. (%), which measures the pruned model performance relative to the dense model across all benchmarks.

**Calibration Dataset.** We construct three different calibration datasets to explore pruning strategies for MLLMs. C4 (Raffel et al., 2020) is a large-scale, general-domain pre-training text corpus, which preserves the model's continuation and generation ability. We follow Wanda (Sun et al., 2023) and randomly select the text segments with a 2048-token length. For Instruction Tuning data (Zheng et al., 2023) and Visual Instruction Tuning (Liu et al., 2023a) data, which enhance instruction following and visual reasoning and understanding abilities, we adopt the default preprocessing pipeline used in LLaVA (Liu et al., 2023a). For calibration strategies: (1) for baseline methods, we follow their default single-type calibration strategies, using 120 samples from C4 randomly. (2) For our MLLM-Pruner, we randomly select 40 samples from each calibration dataset and create a multimodal calibration dataset containing a total of 120 samples. We maximize coverage of the three core capabilities, while ensuring diversity in instruction types, answer lengths, and image complexity (spanning OCR, counting, localization, commonsense QA, and reasoning). In Section A.2 we provide additional detailed analysis on the calibration samples.

**Implementation Details.** For all pruning methods, we focus on unstructured sparsity setting. Following Wanda, our pruning is applied only to the linear layers, while parameters in the image encoder for MLLMs are skipped, as they constitute only a small fraction compared to those in the subsequent language model. The $\beta$ parameter setting is discussed in Section A.1. All pruning and evaluation experiments are performed on NVIDIA A100 GPUs.

| Method | POPE | ScienceQA | TextVQA | MME-Percep. | MME-Cogn. | GQA | MMBench | MMVet | VizWiz | Avg. (%) |
|---|---|---|---|---|---|---|---|---|---|---|
| Dense | 86.5 | 70.4 | 61.3 | 1519.6 | 322.5 | 64.2 | 67.9 | 44.6 | 57.1 | 100.0 |
| Magnitude | 84.9 | 47.1 | 37.4 | 927.7 | 212.9 | 52.7 | 48.4 | 25.1 | 50.6 | 71.2 |
| SparseGPT | 87.0 | **66.9** | **54.0** | 1407.8 | 311.4 | 61.9 | 62.7 | **32.6** | 52.1 | 91.8 |
| Wanda | 86.6 | 64.2 | 52.9 | 1371.6 | 322.1 | 61.7 | 59.9 | 30.5 | 54.35 | 90.6 |
| MLLM-Pruner | **88.4** | 65.6 | 53.5 | **1446.1** | **361.4** | 62.4 | 63.0 | 32.2 | 53.7 | **93.9** |

Table 1: LLaVA-NeXT (Liu et al., 2024a) 7B performance comparison under the 50% sparsity ratio across diverse multimodal evaluation benchmarks. Bold and underlined numbers denote the best and second-best performance, respectively.

| Method | POPE | ScienceQA | TextVQA | MME-Percep. | MME-Cogn. | GQA | MMBench | MMVet | VizWiz | Avg. (%) |
|---|---|---|---|---|---|---|---|---|---|---|
| Dense | 86.3 | 73.5 | 64.3 | 1575.1 | 316.8 | 65.4 | 70.5 | 44.2 | 60.3 | 100.0 |
| Magnitude | 74.4 | 67.6 | 50.5 | 1227.4 | 258.9 | 59.8 | 61.8 | 31.1 | 55.18 | 84.2 |
| SparseGPT | **87.0** | 70.7 | 59.6 | 1518.3 | 277.5 | 63.5 | 64.8 | 37.5 | **51.2** | 92.5 |
| Wanda | 85.1 | 70.2 | 59.5 | 1507.2 | 295.7 | 63.7 | 65.4 | 39.1 | 49.9 | 93.0 |
| MLLM-Pruner | 86.0 | **71.1** | **59.8** | **1530.5** | 296.4 | 63.8 | 65.9 | 41.4 | 50.9 | **94.4** |

Table 2: LLaVA-NeXT (Liu et al., 2024a) 13B performance comparison under the 50% sparsity ratio across diverse multimodal evaluation benchmarks. Bold and underlined numbers denote the best and second-best performance, respectively.

## 4.2 EXPERIMENTAL RESULT

**Pruning Performance for LLaVA Architectures.** As shown in Table 1 and Table 2, we compare our MLLM-Pruner with other representative methods towards LLaVA-NeXT (Liu et al., 2024a) 7B and 13B at a 50% sparsity ratio. For baseline methods, we follow their default single-type calibration dataset C4, and our MLLM-Pruner leverages the proposed multimodal calibration dataset. All experiments are conducted with the same number of calibration samples (120) for a fair comparison. Across nine benchmarks, MLLM-Pruner consistently achieves the best average performance. On LLaVA-NeXT 7B, it achieves an average relative performance of 93.9%, surpassing the strongest baseline (SparseGPT) by 2.1%. On the larger 13B model, MLLM-Pruner reaches 94.4% average relative performance, outperforming Wanda by 1.4%. The results clearly demonstrate the advantage of MLLM-Pruner, which preserves strong multimodal understanding and reasoning capabilities even under high sparsity,

**Robustness to Calibration Sample Size and Sparsity Ratio.**

**(1) Calibration Sample Size.** As illustrated in Fig. 2 a to c, we evaluate the impact of different calibration dataset sizes. We observe that activation-aware methods (Wanda and our MLLM-Pruner) exhibit substantially higher robustness than the iterative-reconstruction based SparseGPT, particularly under limited calibration samples (e.g., 20 samples). Moreover, our MLLM-Pruner consistently outperforms the Wanda baseline across all sample sizes, maintaining the best performance overall. These results demonstrate that our method is largely insensitive to calibration data size, ensuring robustness even with limited samples. **(2) Sparsity Ratio.** As shown in Fig. 2 d, MLLM-Pruner also maintains strong performance across different sparsity levels. Under extremely high sparsity ratios, we observe that SparseGPT performs slightly better than activation-aware methods, benefiting from its more complex iterative process. However, SparseGPT also exhibits greater instability and higher sensitivity to sparsity changes, whereas our approach achieves more consistent and reliable performance across a wide range of sparsity levels.

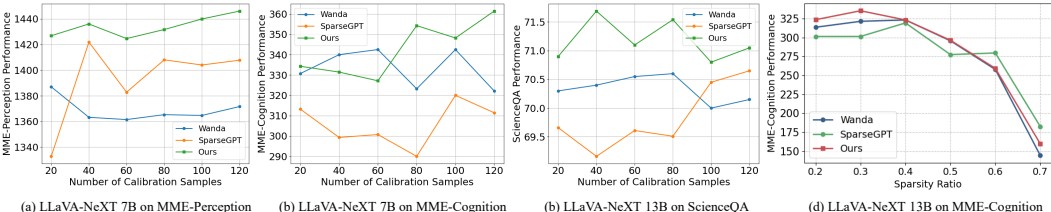

(a) LLaVA-NeXT 7B on MME-Perception  (b) LLaVA-NeXT 7B on MME-Cognition  (b) LLaVA-NeXT 13B on ScienceQA  (d) LLaVA-NeXT 13B on MME-Cognition

Figure 2: Robustness analysis of the calibration sample size and sparsity ratio. Experiments are conducted on LLaVA-NeXT (Liu et al., 2024a) 7B and 13B across multiple benchmarks.

## 4.3 ABLATION STUDY.

MLLM-Pruner differs from previous methods in both the calibration dataset and the activation-aware mode for MLLM. We ablate the two key settings of our MLLM-Pruner to better understand their impact, and compare our method with the activation-aware baseline Wanda (Sun et al., 2023) under the same calibration settings for a fair comparison.

**Effectiveness of the Multimodal Calibration Dataset.** To rigorously evaluate our multimodal calibration strategy, we compare three single-source datasets (C4, Instruction Tuning, and Visual Instruction Tuning) against our proposed hybrid multimodal dataset. Each single-source dataset contains 120 samples, while the hybrid dataset is constructed by sampling 40 instances from each source. As shown in Table 3, existing single-type datasets exhibit specialized strengths but limited generalization. For example, C4 data shows superior performance on text understanding tasks such as TextVQA, while (Visual) Instruction Tuning data excels in vision-oriented intruction-following benchmarks like ScienceQA and MME-Perception. However, these specialized datasets fail to maintain balanced performance across diverse multimodal tasks. In contrast, our hybrid multimodal calibration dataset achieves the highest average performance (95.0%) by strategically combining complementary data sources, and our method outperforms the best single-type calibration by 0.2-

| Method | Type of Calibration Data | POPE | ScienceQA | TextVQA | MME-perception | GQA | Avg. (%) |
|---|---|---|---|---|---|---|---|
| MLLM-Pruner | C4 | **88.9** | 65.2 | **54.1** | 1420.4 | 62.2 | 94.8 |
| | Instruction Tuning | 87.5 | 65.3 | 53.3 | **1457.1** | 62.2 | 94.3 |
| | Visual Instruction Tuning | 88.5 | 65.3 | 53.0 | 1389.8 | 62.2 | 93.6 |
| | Hybrid Data (Ours) | 88.4 | **65.6** | 53.5 | 1446.1 | **62.4** | **95.0** |

Table 3: Ablation study of proposed multimodal calibration dataset for our MLLM-Pruner.

| Method | POPE | ScienceQA | TextVQA | MME-Percep. | MME-Cogn. | GQA | MMBench | MMVet | VizWiz | Avg. (%) |
|---|---|---|---|---|---|---|---|---|---|---|
| Dense | 86.5 | 70.4 | 61.3 | 1519.6 | 322.5 | 64.2 | 67.9 | 44.6 | 57.1 | 100.0 |
| Wanda | 88.2 | 64.4 | 52.4 | 1400.1 | 321.8 | 62.2 | 61.3 | 31.1 | 53.3 | 91.2 |
| Wanda + Ours | **88.4** | **65.6** | **53.5** | **1446.1** | **361.4** | **62.4** | **63.0** | **32.2** | **53.7** | **93.9** |

Table 4: Ablation study of proposed modality-sensitive activation-aware method on LLaVA-NeXT (Liu et al., 2024a) 7B under the 50% sparsity ratio using the same hybird calibration dataset.

1.4% in average accuracy. Notably, it achieves the best performance on ScienceQA (65.6%) and GQA (62.4%), while maintaining competitive results on other benchmarks.

**Effectiveness of Modality-sensitive Activation-aware Pruning.** We further compare our modality-sensitive activation reweighting method with Wanda using the same hybrid calibration dataset. As shown in Table 4, our approach achieves consistent performance improvements across all nine multimodal benchmarks, with an average gain of 2.7% over the baseline, demonstrating our effectiveness in leveraging the attention mechanism for activation-reweighting, which enables more accurate importance estimation and preserving multimodal understanding and reasoning capabilities during MLLM pruning.

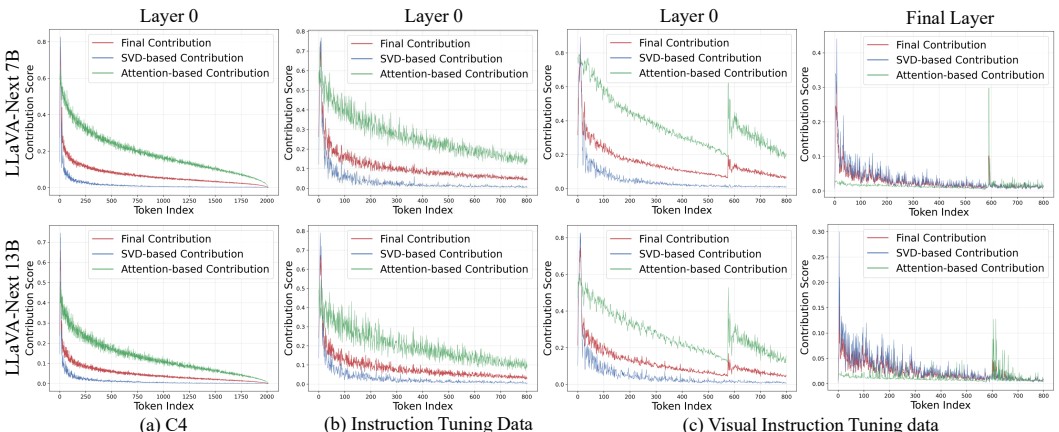

(a) C4  (b) Instruction Tuning Data  (c) Visual Instruction Tuning data

Figure 3: We visualize the differences in attention distributions across three calibration datasets, where each curve represents the average score over all samples. The blue curve corresponds to the mean attention scores across all tokens, while the green curve depicts the SVD-based contribution scores. The impact of our proposed reweighting method is highlighted in red. These distributions are based on LLaVA-NeXT (Liu et al., 2024a) 7B and 13B. More results refer to Appendix A.15

**Visualize Analysis of the Attention Distributions for the Calibration Dataset.** To further investigate MLLM calibration strategies, which remain underexplored in prior work, we visualize the attention distributions in Fig. 3 across three representative datasets: C4, Instruction Tuning (IT), and Visual Instruction Tuning (VIT). Our observations can be summarized as follows: (1) For C4 and IT, the mean attention distribution (green curve) remains relatively smooth, whereas VIT exhibits sharp spikes near token index 576, corresponding to the end of the visual tokens. (2) In VIT, the attention allocated to visual tokens (indices 0–576) progressively decreases from lower to higher layers. However, this decline does not necessarily indicate that visual information becomes less important for pruning. (3) Applying singular value decomposition (SVD) provides a global perspective (blue curve), yielding smoother signals that mitigate both extreme outliers and the apparent down-weighting of visual tokens. By combining attention- and SVD-based contributions, our final contribution score (red curve) enables a more stable and balanced weight-importance estimation.

### 4.3.1 PRUNING SPEED ANALYSIS.

As shown in Table 5, we compare the pruning speed of different methods under the same experimental setup. Specifically, we measure the cumulative time required to prune all layers of the MLLM on an NVIDIA A100 GPU using the same calibration dataset. SparseGPT incurs substantially higher computational over-

| Pruning Model | Wanda | SparseGPT | Ours |
|---|---|---|---|
| LLaVA-NeXT 7B | 47.18 | 465.15 | 50.08 |
| LLaVA-NeXT 13B | 77.57 | 849.22 | 85.83 |

Table 5: Pruning time (seconds) comparison.

head due to the inverse computation. In contrast, Wanda, as a magnitude-based method, is simple and efficient, achieving significantly faster pruning. Our MLLM-Pruner preserves this efficiency, as the activation reweighting step requires only a lightweight single-pass computation per layer. As shown in the last column of the table, our method achieves about $10\times$ speedup over SparseGPT while maintaining competitive pruning performance. Furthermore, as summarized in Table 6, Although our method introduces modality-aware reweighting through activation contribution scores $C_j$, it maintains the same computational complexity as Wanda.

| Method | Weight Update | Calibration Data | Pruning Metric | Complexity |
|---|---|---|---|---|
| Magnitude | ✗ | ✗ | $\lvert W_{ij}\rvert$ | $O(1)$ |
| SparseGPT | ✓ | ✓ | $\lvert W\rvert^2/\mathrm{diag}(XX^T+\lambda I)_{ij}^{-1}$ | $O(d_{\mathrm{hidden}}^3)$ |
| Wanda | ✗ | ✓ | $\lvert W_{ij}\rvert\cdot\lVert X_j\rVert_2$ | $O(d_{\mathrm{hidden}}^2)$ |
| MLLM-Pruner (Ours) | ✗ | ✓ | $\lvert W_{ij}\rvert\cdot\lVert X_j\cdot C_j\rVert_2$ | $O(d_{\mathrm{hidden}}^2)$ |

Table 6: Comparison of pruning methods in terms of weight update, calibration data dependency, pruning metric, and computational complexity.

### 4.3.2 PERFORMANCE FOR QWEN ARCHITECTURE.

To further evaluate the generability of our MLLM-Pruner, we extend experiments to the latest Qwen2.5-VL (Bai et al., 2025) architecture, which supports richer visual input modalities and longer context. We compare MLLM-Pruner with other state-of-the-art baselines under the same multimodal aggregation calibration strategy. As shown in Table 7, MLLM-Pruner consistently outperforms existing methods across various multimodal benchmarks. On average, MLLM-Pruner surpasses the strongest baseline for 1.3% relative improvements.

| Method | TextVQA | DocVQA | OCRBench | MMBench | MMStar | POPE | Avg. (%) |
|---|---|---|---|---|---|---|---|
| Dense | 85.3 | 94.9 | 88.4 | 79.9 | 60.5 | 86.3 | 100.0 |
| Magnitude | 55.8 | 33.3 | 23.8 | 5.5 | 9.3 | 81.1 | 40.6 |
| SparseGPT | 81.7 | **93.7** | 85.1 | 55.9 | 43.1 | 85.3 | 84.9 |
| Wanda | **82.3** | 93.3 | 85.0 | 64.8 | 50.1 | 87.9 | 88.5 |
| MLLM-Pruner | 81.8 | 93.3 | **85.5** | **68.6** | **51.1** | **88.2** | **89.8** |

Table 7: Qwen2.5-VL 7B (Bai et al., 2025) performance comparison under the 50% sparsity ratio across diverse multimodal evaluation benchmarks. Bold and underlined numbers denote the best and second-best performance, respectively.

## 5 CONCLUSION

In this work, we presented **MLLM-Pruner**, an activation-aware post-training pruning framework tailored for multimodal large language models (MLLMs). Our method introduces two key innovations: (1) a multimodal calibration dataset that provides representative and balanced statistics across textual, instructional, and visual-instructional inputs, and (2) a modality-sensitive weight importance estimation metric that explicitly accounts for the multimodal input activation. Extensive experiments on LLaVA-NeXT and Qwen2.5-VL architectures demonstrate that MLLM-Pruner consistently outperforms state-of-the-art baselines, and our method achieves superior trade-offs between pruning efficiency and model performance. Comprehensive visualization analyses and ablation studies provide further insights into the contributions of our method for MLLM pruning.

## 6    ETHICS STATEMENT

Our research fully adheres to the ICLR Code of Ethics, ensuring ethical standards are maintained throughout the whole study.

## 7    REPRODUCIBILITY STATEMENT

To ensure the reproducibility of our research, we have provided comprehensive implementation details, including data construction, model architecture and hyperparameter settings. Additionally, all datasets and data processing steps are fully documented in the supplementary materials. We will also release the complete source code and instructions for reproducing our results.

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

# A  APPENDIX

In this paper, we use Large Language Models to polish writing.

## A.1  ADDITIONAL EXPERIMENTS

**Trade-off Parameter** $\beta$. We further conduct an ablation study to examine the effect of the trade-off parameter $\beta$ in Eq. (8), which balances the attention-based and SVD-based contributions. For a fair comparison, we use the same 120 calibration samples across all settings. As shown in Table 8, our method consistently outperforms the Wanda baseline Sun et al. (2023) for all choices of $\beta$, confirming the effectiveness of modality-sensitive reweighting. In particular, $\beta = 0.0$ (purely SVD-based) and $\beta = 1.0$ (purely attention-based) yield suboptimal but still competitive results, highlighting that combining both signals produces a more reliable and smoother importance estimation. Among the tested values, $\beta = 0.3$ achieves the best overall performance across benchmarks, and we adopt this setting in all subsequent experiments.

| Method | POPE | ScienceQA | TextVQA | MME-Perception | MME-Cognition | GQA | Avg. (%) |
|--------|------|-----------|---------|----------------|---------------|-----|----------|
| Dense | 86.5 | 70.4 | 61.3 | 1519.6 | 322.5 | 64.2 | 100.0 |
| Wanda | 88.2 | 64.4 | 52.4 | 1400.1 | 321.8 | 62.2 | 94.6 |
| Ours ($\beta = 0.0$) | 88.3 | 65.6 | 53.7 | 1426.1 | 341.4 | 62.4 | 96.6 |
| Ours ($\beta = 0.1$) | 88.3 | 65.8 | 53.7 | 1449.1 | 348.9 | 62.4 | 97.3 |
| Ours ($\beta = 0.3$) | 88.4 | 65.6 | 53.5 | 1446.1 | 361.4 | 62.4 | 97.8 |
| Ours ($\beta = 0.5$) | 88.0 | 66.3 | 53.6 | 1433.5 | 345.7 | 62.3 | 97.0 |
| Ours ($\beta = 0.7$) | 88.0 | 66.0 | 54.3 | 1435.7 | 354.6 | 62.4 | 97.5 |
| Ours ($\beta = 0.9$) | 88.2 | 66.3 | 53.9 | 1443.2 | 346.4 | 62.3 | 97.3 |
| Ours ($\beta = 1.0$) | 88.2 | 66.4 | 53.9 | 1435.7 | 313.2 | 62.3 | 95.5 |

Table 8: Ablation study on the trade-off parameter $\beta$ for our modality-sensitive activation-aware pruning method on LLaVA-NeXT (Liu et al., 2024a) 7B under 50% sparsity, using the same hybrid calibration dataset (120 samples).

## Perplexity Evaluation of MLLMs

In this section, we investigate whether perplexity serves as an appropriate evaluation metric for Multimodal Large Language Models (MLLMs). While perplexity on WikiText (Merity et al., 2016) provides a can stably reflect the LLM's performance (Dettmers & Zettlemoyer, 2023), As demonstrated in Table 9, we observe an inverse relationship between text-based perplexity and multimodal performance. The configuration achieving optimal perplexity ($\beta = 1.0$) yields the weakest multimodal results (95.5% average), while our best multimodal performer ($\beta = 0.3$) maintains competitive perplexity (8.89) alongside significantly better multimodal capabilities (97.8% average). This divergence underscores that perplexity, while effective for evaluating LLMs, is inadequate for assessing MLLMs' performance on complex multimodal tasks such as visual reasoning and cross-modal understanding.

| | Wanda | Ours | | | | | | |
|--|-------|------|--|--|--|--|--|--|
| | | $\beta = 0.0$ | $\beta = 0.1$ | $\beta = 0.3$ | $\beta = 0.5$ | $\beta = 0.7$ | $\beta = 0.9$ | $\beta = 1.0$ |
| Wiki Perplexity ↓ | 8.81 | 8.94 | 8.92 | 8.89 | 8.87 | 8.85 | 8.84 | **8.80** |
| Avg. (%) of MLLM Benchmarks ↑ | 94.6 | 96.6 | 97.3 | **97.8** | 97.0 | 97.5 | 97.3 | 95.5 |

Table 9: Comparison between Wanda baseline and our method with different trade-off parameter $\beta$. The discrepancy between Wiki Perplexity and multimodal performance demonstrates the limitation of text-only calibration dataset for MLLM evaluation.

## A.2  DETAILS OF MULTIMODAL CALIBRATION DATASET.

We construct a representative multimodal calibration dataset to preserve continuation, instruction following, and visual understanding capabilities in multimodal large language models (MLLMs). As illustrated in Fig. 4, it consists of three complementary sources: C4 (Raffel et al., 2020), Instruction Tuning data (Zheng et al., 2023), and Visual Instruction Tuning data (Liu et al., 2023a).

As shown in Fig. 5, we visualize the attention distributions of our calibration datasets. The white regions correspond to the causal mask in MLLMs. The training process of MLLMs follows next-token

Figure 4: We illustrate the difference among our calibration sources: C4 (Raffel et al., 2020), Instruction Tuning (Zheng et al., 2023) and Visual Instruction Tuning (Liu et al., 2023a) data.

prediction objective, which is to predict the probability of the upcoming token given all preceding tokens. To prevent information leakage from future tokens, the attention mechanism applies a unidirectional causal mask, which blocks each position from attending to subsequent tokens and restricts the information flow to a left-to-right direction. This property enforces information flow from earlier tokens to subsequent ones, while subsequent tokens remain "invisible" to preceding tokens. As a result, the estimated token importance becomes biased and non-uniform.

As shown in Figs. 6 to 8, we compare the distributions of the Attention-based Contribution and the SVD-based Contribution. In each subfigure, the second column presents the SVD-based contribution scores. From left to right, we demonstrate how the Singular Value Decomposition (SVD) operation progressively smooths the original attention distributions, and the final column represents the final contribution score. This score is computed using Eq. (8), which combines the attention-based and SVD-based metrics through a weighted averaging scheme. The resulting unified metric enables more stable and effective token contribution estimation for pruning, mitigating the biases introduced by the causal attention mechanism while preserving essential multimodal information.

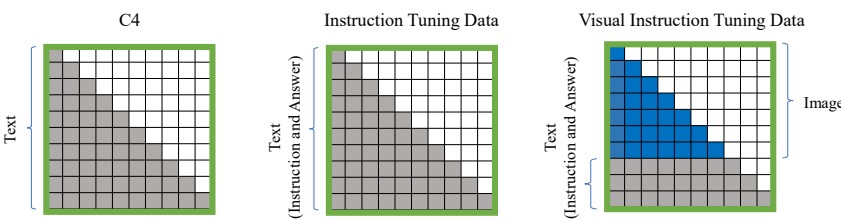

Figure 5: Attention distributions difference for our calibration dataset.

# C4 Calibration Data

## Attention Contribution  SVD Contribution  Final Contribution

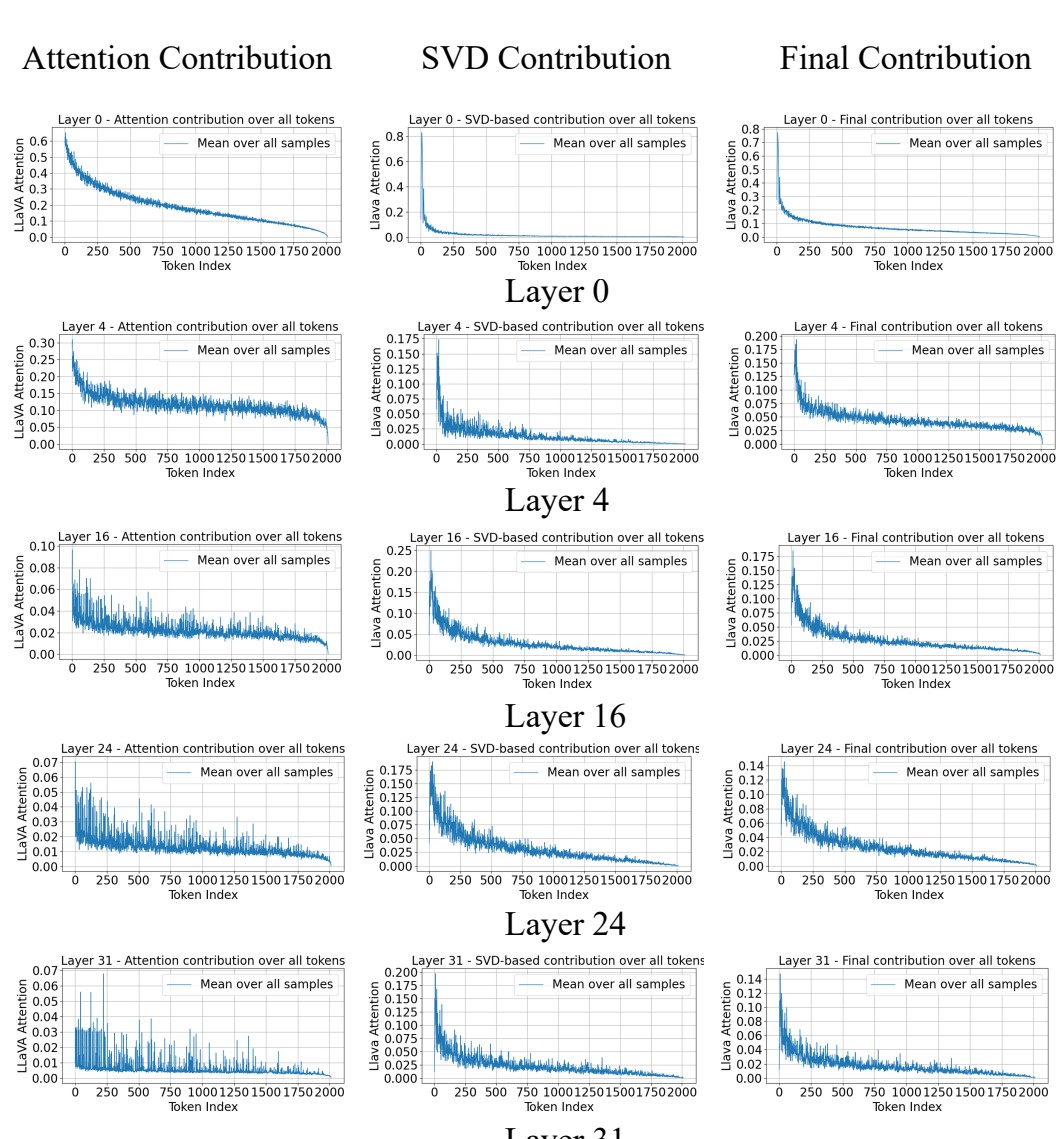

Figure 6: Different contribution score for C4 data.

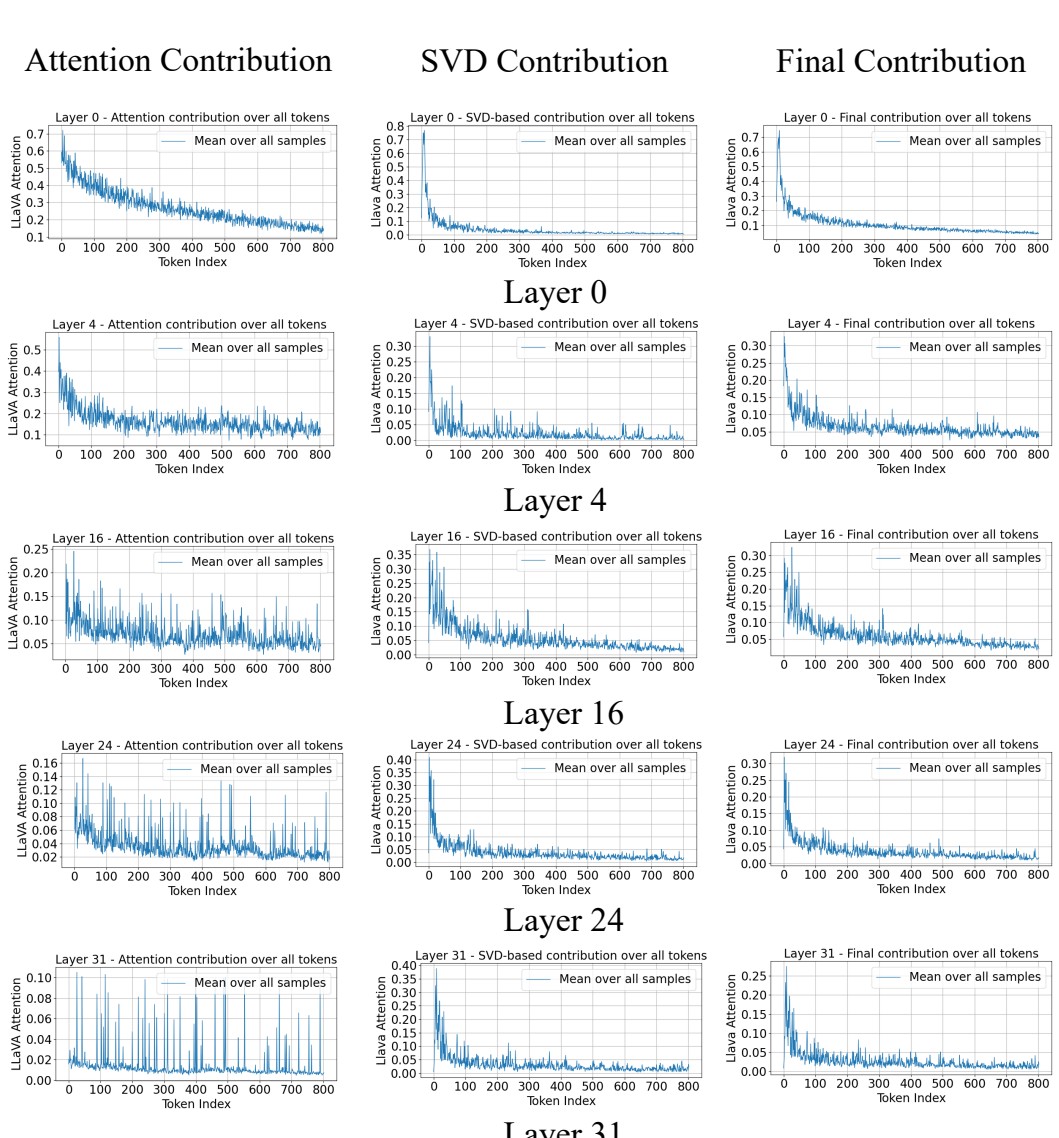

Figure 7: Different contribution score for Instruction Tuning data.

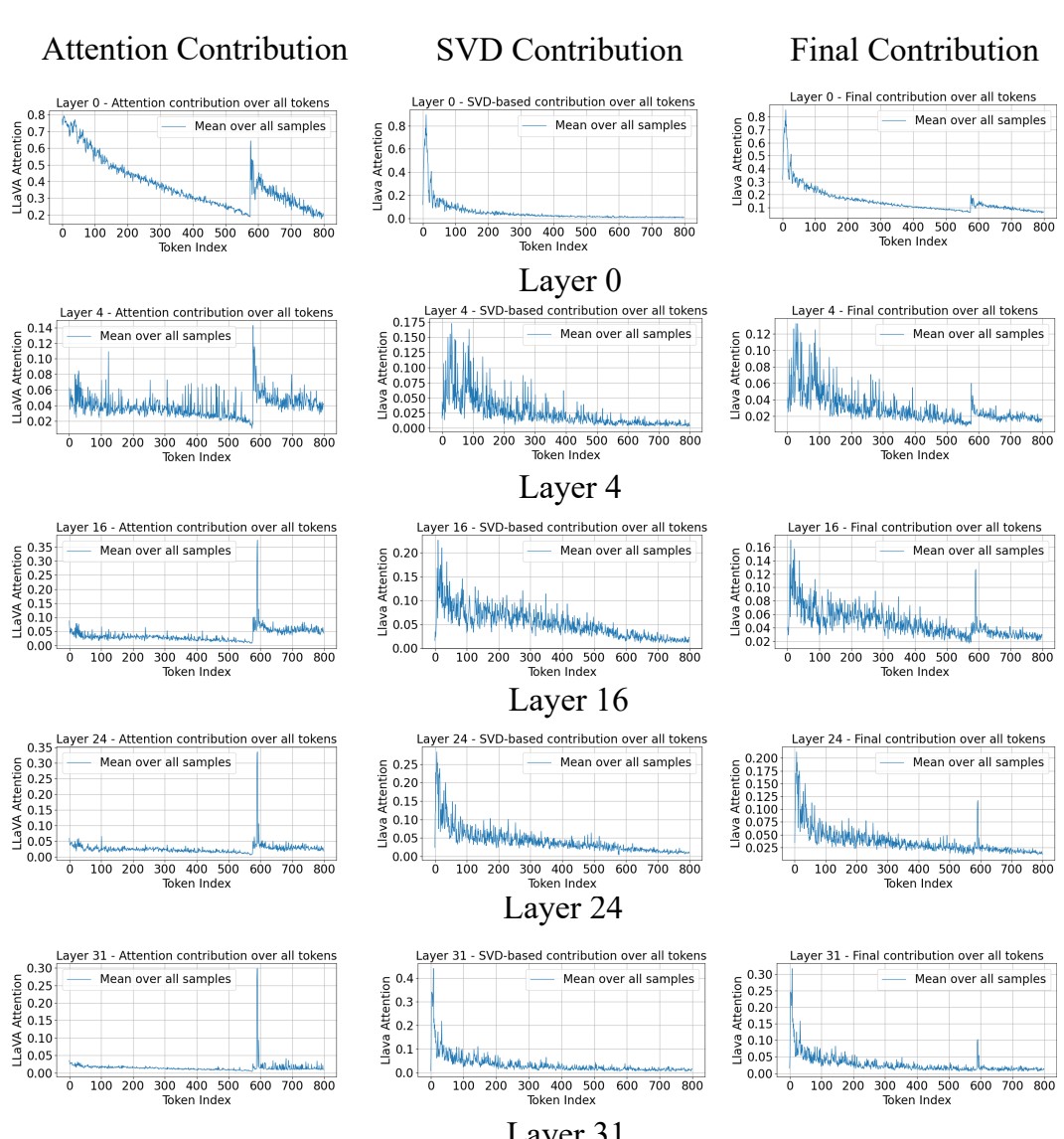

Figure 8: Different contribution score for Visual Instruction Tuning data.

