# OpenReview forum: "MLLM-Pruner: Efficient Activation-aware Pruning for Multimodal LLMs"
_ICLR.cc/2026/Conference — ICLR 2026 Conference Withdrawn Submission_

### Official Review · Reviewer_ZoQX · 2025-10-27

**Soundness:** 1
**Presentation:** 2
**Contribution:** 1
**Rating:** 2
**Confidence:** 4

**Summary:**

The proposed method, MLLM-Pruner, presents two main contributions: a hybrid calibration dataset and a modality-sensitive pruning metric.

**Strengths:**

The MLLM-Pruner achieves performance recovery (outperforming Wanda and SparseGPT on average) while maintaining an $O(D^2)$ time complexity during the pruning process.

**Weaknesses:**

1. Lack of Theoretical Justification for the Modality-Sensitive Metric

The core technical innovation—the modality-sensitive metric—relies on an ad-hoc linear combination of two contribution scores: Attention-based ($\hat{a}^l$) and SVD-based ($\hat{s}^l$), governed by an empirically tuned hyperparameter $\beta$ (found to be $0.3$ in the Appendix).

The authors claim that this combination captures cross-modal divergence, but they provide no theoretical justification for why a simple linear sum, $C^{l}=\beta\hat{a}^{l}+(1-\beta)\hat{s}^{l}$, is the optimal or even a highly effective way to model the complex, non-linear discrepancies between visual and textual token activation flow. The selection of $\beta$ in the appendix, while tested, highlights that the method is sensitive to this arbitrary scaling factor, which undermines the robustness and generality of the proposed metric.

2. Limited Granularity in Modality Reweighting

The proposed $C^l$ score is a token-level reweighting factor (a vector in $\mathbb{R}^{N+M}$), which modulates the input activation $X$ before the $l_2$-norm is computed and multiplied with the weight $W$. This mechanism is uniform across all columns of the weight matrix $W$ within a given layer.

A truly insightful modality-sensitive pruner should explore weight-specific or neuron-specific importance. For instance, some neurons (columns in $W$) might be dedicated to visual processing, while others handle text. The current token-level reweighting does not provide this fine-grained control, which is the necessary step to advance beyond basic activation-aware pruning in the multimodal domain.

3. Marginal and Empirically Fragile Performance Improvement

While MLLM-Pruner outperforms baselines like Wanda and SparseGPT, the average relative performance gains are marginal (e.g., 1.4% to 2.7% relative improvement on LLaVA-NeXT 7B/13B). This marginal gain is achieved at the cost of two extra complexities:
a) Implementing the full SVD decomposition of the attention matrix.
b) Introducing and tuning the critical hyperparameter $\beta$.

For post-training compression, efficiency and simplicity are paramount. The empirical results do not strongly justify the added computational and hyperparameter complexity over the simple and highly efficient Wanda baseline, especially since SparseGPT performs competitively or sometimes better at extreme sparsity ratios (Figure 2d).

4. The Calibration Strategy is an Expected Engineering Baseline

The construction of the hybrid calibration dataset (C4 + IT + VIT) is presented as a major innovation. However, using a combination of text, instruction, and visual instruction tuning data to evaluate an MLLM is arguably the expected engineering baseline for preserving MLLM functionality (continuation, instruction-following, and visual reasoning).

The innovation lies in how one intelligently samples from these sources, not merely combining them. The paper adopts a simple, fixed, and equal split of 40 samples from each source. Deeper analysis should have explored dynamic sampling strategies or loss-based importance weighting across the different data sources, especially since the ablation study (Table 3) shows single-source methods have specialized strengths that are simply averaged out by the hybrid approach.

5. No code

The author didn't provide any implementation detail and code with their submission.

**Questions:**

see above.

---

### Official Review · Reviewer_xsB6 · 2025-10-28

**Soundness:** 3
**Presentation:** 4
**Contribution:** 3
**Rating:** 6
**Confidence:** 4

**Summary:**

This paper identifies limitations of existing activation-aware pruning methods when applied to Multimodal Large Language Models (MLLMs), and proposes MLLM-Pruner, a pruning framework tailored for MLLMs with two core contributions: (1) a multimodal calibration dataset that better matches cross-modal activation patterns, and (2) a modality-sensitive importance estimation that reweights activations using attention and SVD-based contributions before computing weight importance.

**Strengths:**

1. Presents a new post-training, activation-aware pruning framework (MLLM-Pruner) to compress MLLMs and introduces a multimodal calibration dataset aligned with multimodal activation characteristics.

2. Provides extensive experiments across multiple models and benchmarks, demonstrating the method’s effectiveness.

**Weaknesses:**

1. Lack of theoretical grounding for the SVD metric. The paper does not provide a formal derivation to justify why the SVD-based contribution is theoretically well-founded; it reads more like a practically effective heuristic.

2. Potential overlap with related work (TokenCarve). TokenCarve (Tan et al., 2025) also leverages SVD-based contribution combined with attention (for visual token compression). While MLLM-Pruner applies SVD+attention to activation reweighting for weight pruning, the final activation-contribution formulation (e.g., Eq. 8) appears highly similar in spirit. Please clarify precise differences.

3. Generalization of the hyperparameter \beta. Table 8 reports \beta=0.3 as best and then adopts it for subsequent experiments. Did you validate whether \beta=0.3 remains optimal for other architectures (e.g., Qwen2.5-VL)? If yes, please provide evidence; if not, discuss sensitivity and whether per-model tuning is required.

Tan X, Ye P, Tu C, et al. Tokencarve: Information-preserving visual token compression in multimodal large language models[J]. arXiv preprint arXiv:2503.10501, 2025.

**Questions:**

Please refer to the 'weaknesses' .

---

### Official Review · Reviewer_Ayii · 2025-11-01

**Soundness:** 2
**Presentation:** 3
**Contribution:** 2
**Rating:** 2
**Confidence:** 3

**Summary:**

## Summary

This paper introduces MLLM-PRUNER, an efficient post-training pruning framework that adapts an **activation-aware** $\ell_2$-norm metric to multimodal large language models (MLLMs). By leveraging a small calibration dataset to estimate activation importance, the method aims to reduce model size and theoretical FLOPs with minimal performance loss on various VQA benchmarks.

**Strengths:**

## Strengths

1.  **S1: Adapting Activation-Aware Pruning for Multimodal LLMs.**
    The paper successfully adapts an effective activation-aware pruning metric—combining weight magnitude and input activation $\ell_2$-norm—from the unimodal LLM domain to the complex MLLM structure. This approach represents a principled effort to move beyond simple magnitude pruning by incorporating runtime activation information.

2.  **S2: Post-Training Efficiency and Broad Model Coverage.**
    MLLM-PRUNER is a post-training method, which avoids costly fine-tuning. The authors demonstrate its effectiveness across a relatively broad spectrum of modern MLLM architectures (like LLaVA and mPLUG-Owl), showcasing its potential as a plug-and-play tool for efficient MLLM deployment.

**Weaknesses:**

## Weaknesses

1.  **W1: Lack of Theoretical Basis and Multimodal Inapplicability of the Core Metric.**
    The paper's core approach is simply porting the $\ell_2$-norm activation-aware metric from unimodal LLMs. This strategy *fundamentally ignores two core challenges* of multimodal pruning: a) the vast difference in activation distributions between vision and text tokens; and b) the specific fragility of cross-modal attention post-pruning. The lack of any justification that this simple $\ell_2$ combination captures the *critical information alignment* specific to multimodal fusion *fundamentally challenges* the metric's validity and effectiveness.

2.  **W2: Calibration Overhead Contradicts Post-Training Efficiency Claim.**
    MLLM-Pruner relies on a calibration dataset to obtain activation $\ell_2$-norms. For large MLLMs, even a single calibration pass can incur *significant computational and time cost*.
    * If the overhead of this "calibration" process substantially exceeds the cost of minimal fine-tuning (e.g., via LoRA), the method *loses its appeal* as an "efficient" post-training pruning technique.
    * The paper fails to provide the time and resource cost required for calibration, leaving its efficiency claim *unsubstantiated*.

3.  **W3: Insufficient Performance Evaluation and Missing Multiturn Dialogue.**
    The paper relies on a limited subset of VQA tasks for evaluation, which *fails to fully reflect* real-world MLLM performance.
    * *Missing* is the robustness analysis on challenging benchmarks like Multiturn Conversation and complex visual reasoning.
    * Since these tasks are far more sensitive to pruning-induced errors, the lack of this crucial validation limits confidence in MLLM-Pruner's *generalizability and robustness* in practical applications.

**Questions:**

## Questions

1.  **Q1: Multimodal Justification for the Core Metric.**
    The activation-aware metric in MLLM-Pruner is directly ported from **unimodal LLMs**. Visual and text tokens have a **vast difference** in activation distribution and feature dimension. How can the authors prove, from a **theoretical or information-theoretic** standpoint, that a simple combination of the $\ell_2$-norms of these two modalities accurately captures the critical information that influences the **fragility of multimodal fusion** within the **cross-attention layer**? Without this proof, is the application of this metric in MLLMs merely **heuristic** rather than principled?

2.  **Q2: Efficiency and Overhead of the "Calibration Dataset."**
    The paper claims to be an efficient **post-training pruning** method. However, the overhead of using a **calibration dataset** to perform a full network pass for $\ell_2$-norm estimation is **not quantified**. How does the time and resource cost (GPU hours) required for this "calibration" process compare to the overhead of performing **lightweight fine-tuning** (e.g., via LoRA) with a simple magnitude-based pruning method? If the calibration overhead is higher, how does MLLM-Pruner justify its superiority in **practical efficiency**?

3.  **Q3: Completeness of Performance Evaluation and Missing Multiturn Dialogue.**
    The robustness of pruned MLLMs on **complex tasks** is crucial. The paper's evaluation is primarily limited to a subset of VQA tasks. Can the authors provide performance data on **Multiturn Conversation** datasets (e.g., VisDial or related benchmarks)? Given that contextual dependency and accumulated visual redundancy in these tasks are highly sensitive to pruning, doesn't the lack of this validation imply that MLLM-Pruner **cannot guarantee** its robustness in practical dialogue systems?

4.  **Q4: Disconnect between Efficiency Claims and Real Latency Data.**
    The main evidence for efficiency is the **theoretical reduction in FLOPs**. However, for this **Unstructured Pruning**, theoretical FLOPs reduction often does not translate directly to **actual latency speedup**. Can the authors provide quantified data on the **real End-to-End Latency Speedup** during inference on standard hardware (e.g., V100 or A100)? If real latency data is missing, doesn't the **"efficient"** claim lack essential hardware-level support?


## Suggestions

1.  **G1: Strengthen the Theoretical Foundation of the Multimodal Metric.**
    Given that MLLM-Pruner's core metric is ported directly from unimodal LLMs, the authors should attempt to **validate** or **refine** the $\ell_2$-norm combination using **causal or information-theoretic methods** (e.g., inspiration from Transfer Entropy or PID). This can be achieved via an **ablation study** to prove the metric's superior \textbf{sensitivity to cross-modal interactions} compared to simple weight magnitude or pure $\ell_2$ activation, thereby grounding its **principled application** in MLLMs.

2.  **G2: Quantify and Mitigate the Real Overhead of the Calibration Process.**
    To fully substantiate MLLM-Pruner's efficiency claim, the authors **must** quantify the time and resource cost (e.g., GPU hours on an A100) required for the **calibration dataset pass**. Furthermore, the authors should investigate the robustness of using **smaller, domain-agnostic** data subsets for calibration, or explore ways to **batch or approximate** the $\ell_2$-norm estimation to minimize the impact of calibration on deployment efficiency.

3.  **G3: Adopt Fixed Token Retention Rates for Intuitive Comparison and Update Baselines.**
    To provide a more intuitive and persuasive comparison of pruning efficacy, it is recommended that the experiments include performance data at **fixed token retention rates** (e.g., 11.1% or 22.2% visual tokens, or 64/128 tokens for LLaVANext), and compare against **advanced token pruning methods** like VisionZip or VSCAN. This presentation style, commonly used in prior token pruning works, would more directly demonstrate MLLM-Pruner's **optimal performance advantage** at specific, relevant pruning rates, and address the issue of using older baselines.

---

### Official Review · Reviewer_vtxZ · 2025-11-01

**Soundness:** 2
**Presentation:** 2
**Contribution:** 2
**Rating:** 2
**Confidence:** 4

**Summary:**

Overall, the paper proposes an efficient activation-aware pruning method for MLLMs, but its core methodological contribution is severely undermined by a lack of rigorous theoretical justification for the combined SVD/Attention metric, insufficient comparison against recent baselines, and a critically incomplete evaluation on essential multi-turn dialogue and generative robustness tasks.

**Strengths:**

1.  The paper clearly identifies the critical limitation of applying traditional activation-aware pruning methods to MLLMs, namely the failure to account for the distinct activation patterns of vision and text tokens.

2.  MLLM-Pruner remains a non-iterative, post-training pruning method, retaining the computational efficiency necessary for practical deployment.

**Weaknesses:**

1. The experimental validation is weak due to a limited and outdated set of baselines, failing to place MLLM-Pruner within the current state-of-the-art methods for MLLM pruning. The paper critically omits comparisons with recent, specialized MLLM pruning methods, including **VisionZip**, **SparseVLM**, or **PyramidDrop**. The paper cannot claim to offer a SOTA solution for MLLM pruning without demonstrating superiority over these direct competitors. Outperforming older, generalized LLM pruning methods is insufficient for acceptance.

2. The proposed metric $C^l = (1-\beta) \hat{a}^l + \beta \hat{s}^l$ is a highly heuristic linear combination. The authors provide no rigorous mathematical or mechanistic explanation for why the SVD component ($\hat{s}^l$) effectively captures "multimodal differences" or is fundamentally necessary for weight importance estimation. It appears to be an **ad-hoc heuristic patch**. Introducing SVD adds computational overhead and the hyperparameter $\beta$. If its contribution to performance is marginal (as hypothesized in W1), it unjustifiably increases the method's **complexity, cost, and tuning difficulty**, undermining its claimed efficiency.

3. The evaluation is incomplete, focusing primarily on single-turn, factual visual question-answering tasks (VQA, GQA). The core application of modern MLLMs is complex, long-context, **multi-turn dialogue**. Pruning can severely degrade a model's ability to maintain context and historical memory. By **omitting testing on multi-turn dialogue benchmarks**, the authors fail to demonstrate the practical robustness and real-world applicability of MLLM-Pruner.

4. While the paper introduces a multimodal calibration set (combining C4, IT, and VIT data) to ensure comprehensive representativeness, the strategy for determining the mixing proportions ($\alpha_d$) of these heterogeneous data sources lacks rigorous experimental or theoretical justification. This absence of analysis on the **statistical properties** of the final blended calibration set (e.g., whether it uniformly covers high-activation regions under both visual and linguistic inputs) is a critical methodological flaw. If the blending ratio is merely heuristic or empirically determined, it introduces a significant risk of **calibration set distribution bias** or **overfitting** to the specific test set distribution. Consequently, the performance gains achieved might be an artifact of this specific data composition rather than the inherent generalizability of the pruning metric. This lack of robustness verification prevents the method from being considered an architecture-agnostic pruning framework suitable for diverse MLLM models and tasks.

**Questions:**

1.  Please supplement the results with performance metrics on established **multi-turn dialogue** datasets. The impact of pruning on **long-context memory** must be quantified.

2.  Provide quantitative evidence demonstrating how the SVD component ($\hat{s}^l$) specifically and non-trivially up-weights the importance scores of **visual tokens** or **cross-modal interaction tokens**. Provide a comparison of the $\hat{s}^l$ distribution for image-present vs. text-only inputs.

3.  To demonstrate MLLM-Pruner's true SOTA competitiveness in MLLM pruning, **please supplement the experiments with detailed comparisons against recent and popular MLLM pruning methods (e.g., VisionZip, SparseVLM, VSCAN, PyramidDrop, etc.)**. Furthermore, please validate the effectiveness and generalization ability of MLLM-Pruner **on a more diverse set of MLLM architectures (e.g., InstructBLIP, InternVL3 etc.)**. Results solely on the LLaVA(only llava1.6) and Qwenvl(only 2.5) series are insufficient to prove the method's architecture-agnostic nature.

4.  Provide a detailed sensitivity curve for the hyperparameter $\beta$ across **all individual downstream tasks** . Is there a single, universal $\beta$ value that performs optimally across all tasks and sparsity ratios?

5.  Given that the performance of activation-aware pruning is fundamentally tied to the **statistical representativeness** of the calibration set, the reliance on an empirically determined mixing ratio for heterogeneous data (C4/IT/VIT) introduces significant doubt. Could the authors provide a rigorous **quantitative analysis**—for instance, using **Activation Entropy** or **Principal Component Analysis (PCA) on the activation space**—to prove that the final blended calibration set **statistically covers** the full distribution of high-importance activations across all necessary multimodal tasks? If this mixing ratio is found to be sub-optimal or requires re-tuning when migrating the framework to a different MLLM architecture (e.g., InternVL3), how can the method be classified as an architecture-agnostic pruning framework, and what is the scientific guidance for setting this critical ratio?

---

### Note · Authors · 2025-11-18

I have read and agree with the venue's withdrawal policy on behalf of myself and my co-authors.